# Chemical Fingerprinting and Biological Evaluation of the Endemic Chilean Fruit *Greigia sphacelata* (Ruiz and Pav.) Regel (Bromeliaceae) by UHPLC-PDA-Orbitrap-Mass Spectrometry

**DOI:** 10.3390/molecules25163750

**Published:** 2020-08-17

**Authors:** Ruth E. Barrientos, Shakeel Ahmed, Carmen Cortés, Carlos Fernández-Galleguillos, Javier Romero-Parra, Mario J. Simirgiotis, Javier Echeverría

**Affiliations:** 1Instituto de Farmacia, Facultad de Ciencias, Universidad Austral de Chile, Valdivia 5090000, Chile; ruth.barrientos@alumnos.uach.cl (R.E.B.); shakeel.ahmed@uach.cl (S.A.); carmenc1012@gmail.com (C.C.); carlos.fernandez@uach.cl (C.F.-G.); 2Departamento de Química Orgánica y Fisicoquímica, Facultad de Ciencias Químicas y Farmacéuticas, Universidad de Chile, Olivos 1007, Casilla 233, Santiago 8380544, Chile; javier.romero@ciq.uchile.cl; 3Departamento de Ciencias del Ambiente, Facultad de Química y Biología, Universidad de Santiago de Chile, Casilla 40, Correo 33, Santiago 9170002, Chile

**Keywords:** *Greigia sphacelata*, endemic fruits, UHPLC-PDA-Orbitrap-MS, metabolomic analysis, antioxidant, cholinesterase inhibition

## Abstract

*Greigia sphacelata* (Ruiz and Pav.) Regel (Bromeliaceae) is a Chilean endemic plant popularly known as “quiscal” and produces an edible fruit consumed by the local Mapuche communities named as “chupón”. In this study, several metabolites including phenolic acids, organic acids, sugar derivatives, catechins, proanthocyanidins, fatty acids, iridoids, coumarins, benzophenone, flavonoids, and terpenes were identified in *G. sphacelata* fruits using ultrahigh performance liquid chromatography-photodiode array detection coupled with a Orbitrap mass spectrometry (UHPLC-PDA-Orbitrap-MS) analysis for the first time. The fruits showed moderate antioxidant capacities (i.e., 487.11 ± 26.22 μmol TE/g dry weight) in the stable radical DPPH assay, 169.08 ± 9.81 TE/g dry weight in the ferric reducing power assay, 190.32 ± 6.23 TE/g dry weight in the ABTS assay, and 76.46 ± 3.18% inhibition in the superoxide anion scavenging assay. The cholinesterase inhibitory potential was evaluated against acetylcholinesterase (AChE) and butyrylcholinesterase (BChE). From the findings, promising results were observed for pulp and seeds. Our findings suggest that *G. sphacelata* fruits are a rich source of diverse secondary metabolites with antioxidant capacities. In addition, the inhibitory effects against AChE and BChE suggest that natural products or food supplements derived from *G. sphacelata* fruits are of interest for their neuroprotective potential.

## 1. Introduction

Plant-based foods, especially fruits, have a great impact on human health [1,2]. Indeed, there is a growing body of scientific literature describing curative effects of fruits against a broad spectrum of diseases including diabetes, obesity, neurodegenerative, gastric and cardiovascular disorders, and certain types of cancers [3,4,5,6]. Different types of secondary metabolites have been reported in fruits [4,7,8,9]. These compounds are in part responsible for their biological properties.

The genus *Greigia* (Bromeliaceae) comprise approximately 36 species in Central and South America and form an extraordinary disjunct distribution in humid habitats. Among Chilean *Greigia*, four endemic species have been described: *G. pearcei* (Mez), *G. landbeckiia* (Lechl. ex Phil.) F.Phil., *G. berteroi* (Skottsb), and *G sphacelata* (Ruiz and Pav.) Regel. *Greigia sphacelata* (Ruiz and Pav.) Regel (Bromeliaceae) (local name: Chupon and quiscal) is an endemic plant (Figure 1a,b) widely distributed in temperate zones of Central and Southern Chile [10].

The edible fruits of *G. sphacelata* are juicy, sweet, and with a similar flavor to apple and pineapple, and are eaten raw, or prepared as an infusion or as a fermented beverage. Few reports have investigated the chemical properties of *G. sphacelata*. From the aerial parts of *G. sphacelata* flavanones (5,7,3′-trihydroxy-6, 4′,5′-trimethoxyflavanone, and 5,3′-dihydroxy-6,7,4′,5′-tetramethoxyflavanone), glycerol derivatives (1-*O*-*trans*-*p*-coumaroylglycerol, 1,3-*O*-di-*trans*-*p*-coumaroylglycerol, 1-(ω-feruloyl-docosanoyl) glycerol, and 1-(ω-feruloyl-tetracosanoyl) glycerol), and *trans*-ferulic acid, arborinone, 22-hydroxydocosanoic acid ester, isoarborinol and arborinol have been isolated and characterized [11]. To our knowledge, there are no reports on the chemical characterization or antioxidant and enzyme inhibitory properties of the natural products obtained from *G. sphacelata* fruits.

HPLC or UHPLC coupled to mass spectrometry is a key technique for plant chemotaxonomy or comparative metabolic profiling. Q-Exactive type focus equipment uses a very rapid high-performance mass spectrometer for the detection of small organic molecules [12,13,14]. This is a dual high resolution with accurate mass (HRAM) spectrometer with an orbital trap (Orbitrap), a quadrupole (Q), and a high-performance collision cell (HCD), capable of producing high-resolution parent ions and diagnostic MS daughter fragments. The hyphenated ultrahigh performance liquid chromatography-photodiode array detection coupled with a Orbitrap mass spectrometry (UHPLC-PDA-Orbitrap-MS) approach is a key tool for the identification of secondary metabolites in plants and edible fruits [15,16,17].

In this work, we report the antioxidant activity, cholinesterase inhibitory potential plus the UHPLC-PDA-Orbitrap-MS fingerprinting from the endemic *G. sphacelata* fruits for the first time.

## 2. Results and Discussion

### 2.1. Metabolomic Analyses

In this study, the fingerprint was generated using UHPLC-PDA-Orbitrap-MS (Figure 2) allowing the determination of several types of metabolites in the fruits of the Mapuche specie *G. sphacelata* (Table 1). The metabolites identified include: Eleven phenolic acids (peaks **1**, **6**, **8**, **13**, **14**, **16**, **21**, **42**, **47**, **48**, and **54**); six organic acids (peaks **2**–**5**, **7**, and **10**, including a vitamin peak **7**); eleven sugar derivatives (peaks **12**, **15**, **19**, **22**–**24**, **26**, **31**, **39**, **40**, and **59**); five catechins and/or proanthocyanidins (peaks **34**, **37**, **41**, **44**, and **49**); thirteen fatty acids, (peaks **33**, **50**, **52**, **55**, **56**, **58**, **62**–**67**, and **69**); four iridoids (peaks **11**, **17**, **18**, and **68**); three coumarins (peaks **25**, **27**, and **30**); one benzophenone (peak **57**); ten flavonoids (peaks **20**, **28**, **29**, **32**, **35**, **36**, **38**, and **43**–**46**); five terpenes, (peaks **9**, **51**, **53**, **60**, and **61**).

Examples of full MS spectra and structures of the compounds identified are displayed in Appendix A. The detailed identification is explained below:

#### 2.1.1. Coumarins

Several metabolites were identified as coumarins, due to their characteristic UV_max_ spectra. Peak **25** with an anion [M − H]^−^ (molecule without an hydrogen, charged negatively, formed by the heated electrospray device) at *m/z* (mass/charge ratio) 353.05130 was identified as aesculetin-7-*O*-glucuronide (C_15_H_13_O_10_^−^) and peak **27** with a [M − H]^−^ ion at *m/z* 337.05643 was identified as 7-hydroxycoumarin-glucuronide (C_15_H_13_O_9_^−^). Peak **30**, with a [M − H]^−^ ion at *m/z* 367.06696 scopoletin 7-*O*-glucuronide (C_16_H_17_O_10_^−^).

#### 2.1.2. Flavonoids

Peak **29** with a UV_max_ at 281 nm and with a parent ion at *m/z* 461.10894 was determined as tectoridin (C_22_H_21_O_11_^−^). Peak **32** with a similar UV_max_ and a MS ion at *m/z* 351.08591 was named as lupinisoflavone (C_20_H_15_O_6_^−^). Peak **35** with an anion [M − H]^−^ at *m/z* 593.15024 was determined as genistein-7-*O*-di-glucoside (C_27_H_29_O_15_^−^) producing a genistein aglycone ion at *m/z* 269.0452 and peak **36** as a di-galactoside derivative. Peak **38** with a pseudo-molecular ion at *m/z* 431.09824 was named as genistein-7-*O*-glucoside (C_21_H_19_O_10_^−^) which produced a MS^2^ daughter ion at *m/z* 269.0452 (genistein). Peaks **43** and **45** with ions around *m/z* 415.10364 were identified as the isomers: Daidzein-7-*O*-galactoside and glucoside (C_21_H_19_O_9_^−^, respectively). Finally, peak **46** was named as ononin (formononetin 7-*O*-glucoside (C_22_H_21_O_9_^−^). Peak **20** with an anion [M − H]^−^ at *m/z* 533.16435 was determined as amurensin (C_26_H_29_O_12_^−^), the tert-amyl alcoholic derivative of the flavonol kaempferol 7-*O*-glucoside, while peak **28** with a pseudo-molecular anion at *m/z* 505.09767 was determined as the related flavonol quercetin-3-*O*-glucoside-acetate (C_23_H_21_O_13_^−^).

#### 2.1.3. Hydrocarbons and Saturated Organic Acids

Peak **2** was identified as quinic acid (C_7_H_11_O_6_^−^), Peaks **3** and **5** with ions around *m/z* 191.01933 were determined as citric acid and isocitric acid (C_6_H_7_O_7_^−^), respectively. Peak **4** with an ion [M − H]^−^ at *m/z* 173.00879 was named as dehydroascorbic acid (C_6_H_5_O_6_^−^), while peak **7** with a [M − H]^−^ ion at *m/z* 205.03492 was labeled as homocitric acid (C_10_H_17_O_4_^−^). Peak **10** was identified as pantothenic acid.

#### 2.1.4. Oxylipins or Fatty Acids

Several metabolites were identified as polyhydroxylated unsaturated fatty acids known as the dietary antioxidants oxylipins. Accordingly, peak **33** was identified as 2,2,3-Tris(2,3-dihydroxypropanoyl) decanoic acid (C_19_H_31_O_11_^−^), peak **50** with an anion [M − H]^−^ at *m/z* 287.15001 was determined as a tetrahydroxy-tetradecadienoic acid (C_14_H_23_O_6_^−^), peak **52** with a [M − H]^−^ ion at *m/z* 355.17648 was determined as pentahydroxy-octadecatetraenoic acid (C_18_H_27_O_7_^−^), peak **55** with a [M − H]^−^ ion at *m/z* 273.20734 was named as dihydroxy-pentadecanoic acid (C_15_H_29_O_4_^−^), peak **56** as 12-hydroxyoctadecanoic acid (C_18_H_36_O_3_^−^) and peak **58** as its monounsaturated derivative hydroxy-pentadecaenoic acid ion at *m/z* 271.19173 (C_15_H_27_O_4_^−^). Furthermore, peak **62** with an anion [M − H]^−^ at *m/z* 313.23734 was reported as dihydroxy-octadecaenoic acid (C_18_H_33_O_4_^−^), peak **63** as pentadecanedioic acid (C_15_H_27_O_4_^−^) while peak **64** (ion at *m/z* 327.21780) was identified as trihydroxy-octadecadienoic acid (C_18_H_31_O_5_^−^), and peak **65** as trihydroxy-octadecaenoic acid while the related compound peak **67** with an ion at *m/z* 327.18142 was ascribed to tetrahydroxy-heptadecatrienoic acid (C_17_H_27_O_6_^−^). Peak **66** with an anion [M − H]^−^ at *m/z* 273.20724 was identified as 2,3-dihydroxypropyl dodecanoate (C_15_H_29_O_4_^−^). Peak **65** with an anion [M − H]^−^ at *m/z* 329.23341 was assigned to trihydroxy-octadecaenoic acid (C_18_H_33_O_5_^−^) as already reported by some of us from the fruits of the *Gomortega keule* (Molina) Baill. (Keule tree). Finally, peak **69** with an anion [M − H]^−^ at *m/z* 257.21222 was assigned to hydroxy-pentadecanoic acid (C_15_H_29_O_3_^−^) and peak **70** unknown compound matches formula (C_14_H_29_O_8_^−^).

#### 2.1.5. Iridoids

Three compounds were identified as iridoids, these compounds are very important because they have exhibited a wide range of bioactive effects, such as hypolipidemic, antioxidant, antimicrobic, hypoglycaemic, choleretic, antispasmodic, besides immuno-modulatory anti-inflammatory, neuroprotective, hepatoprotective, and cardioprotective effects [18]. Peak **11** with an ion [M − H]^−^ at *m/z* 443.19117 was determined as the iridoid ebuloside (C_21_H_31_O_10_^−^). Peak **68** with an anion [M − H]^−^ at *m/z* 441.21329 was identified as the acylated iridoid jatamanvaltrate H (C_22_H_33_O_9_^−^). Peak **17** with an anion [M − H]^−^ at *m/z* 491.15479 was determined as cinnamoyl catalpol and peak **18** as its isomer (C_24_H_27_O_11_^−^).

#### 2.1.6. Terpenes

Several compounds were ascribed as having the terpenoid (C-20 or C-30) skeleton. Peak **9** with an anion [M − H]^−^ at *m/z* 365.14422 was tentatively determined as euonyminol (C_15_H_25_O_10_^−^). Peak **51** with a pseudo-molecular ion at *m/z* 343.17513 was identified as the terpenoid monic acid (C_17_H_27_O_7_^−^) and peak **53** as the sesquiterpene dictamnoside N [19]. Peaks **60** and **61** with ions at *m/z* 331.19039 and 485.32615 were identified as the diterpene marrubiin (C_20_H_27_O_4_^−^) and triterpene quillaic acid (C_30_H_45_O_5_^−^), respectively.

#### 2.1.7. Benzophenones

Peak **57** with a pseudo-molecular anion at *m/z* 431.22061 was identified as the dibenzophenone derivative congestiflorone (C_28_H_31_O_4_^−^).

#### 2.1.8. Phenolic Acids and Derivatives

Some eleven compounds were characterized as phenolic acids and derivatives. Peak **1** with an anion [M − H]^−^ at *m/z* 377.08671 was determined as syringaldehyde syringate (C_18_H_17_O_9_^−^), peak **6** as diffutidin (C_17_H_18_O_5_^−^), Peak **8** with a [M − H]^−^ ion at *m/z* 329.08671 was labeled as vanilloyl-glucoside (C_14_H_17_O_9_^−^), while peak **13** with a [M − H]^−^ ion at *m/z* 359.09824 was assigned as glucosyringic acid (C_15_H_19_O_10_^−^). Peak **14** as acetylshikonin (C_18_H_17_O_6_^−^), peak **16** with a parent ion at *m/z* 385.07654 was assigned as *O*-feruloyl-galactarate (C_16_H_17_O_11_^−^). Peak **21** as 2-caffeoylisocitric acid (C_15_H_13_O_10_^−^), peak **42** with a [M − H]^−^ ion at *m/z* 253.07066 was determined as bis-2-hydroxyethyl phtalate (C_12_H_13_O_6_^−^). Some two compounds were identified as phenolic acid glycerol derivatives. Peak **47** with an anion [M − H]^−^ at *m/z* 253.07150 was resolved as 1-*O*-*trans*-*p*-coumaroylglycerol (C_12_H_13_O_6_^−^) while peak **48** with an anion [M − H]^−^ at *m/z* 415.10236 was named as 1,-*O*-di-*trans*-*p*-coumaroylglycerol (C_21_H_19_O_9_^−^) and peak **54** as evodinnol (C_14_H_16_O_4_^−^).

#### 2.1.9. Catechins and Proanthocyanidins

Peak **49** with a parent ion at *m/z* 289.07164 was determined as the flavanol catechin (C_15_H_13_O_6_^−^), peak **34** with a parent ion at *m/z* 865.19451 was resolved as a trimeric procyanidin (C_45_H_37_O_18_^−^), the fragmentation pattern is in concordance for the trimeric procyanidin C2, (C_45_H_37_O_18_^−^) isolated from *Pinus radiata* D. Don. Peak **37** with an ion [M − H]^−^ at *m/z* 577.13512 was identified as procyanidin B1 (C_30_H_23_O_12_^−^), while peak **41** with a parent ion [M − H]^−^ at *m/z* 1153.25147 was identified as proanthocyanin tetramer (C_60_H_49_O_24_^−^) and finally, peak **44** was resolved as procyanidin A1 (C_30_H_23_O_12_^−^).

#### 2.1.10. Sugar Derivatives

The most abundant compounds in *G. sphacelata* fruits were the sugar derivatives. Peak **12** with an ion [M − H]^−^ at *m/z* 409.17191 was resolved as the sugar derivative nonioside K (C_17_H_29_O_11_^−^), while peak **15** with a parent ion [M − H]^−^ at *m/z* 381.13914 was determined as allyl-sucrose (C_15_H_25_O_11_^−^). Peak **22** with a [M − H]^−^ ion at *m/z* 479.17749 was resolved as methyl 2,3,4-tris-*O*-[2-(carboxymethyl) ethyl]-glucose derivative (C_26_H_31_O_13_^−^) and peak **23** as methyl 2,3,4-tris-*O*-[2-(carboxymethyl) ethyl]-glucose (C_19_H_31_O_12_^−^). Peak **24** as (2*E*)-2,6-dimethyl-6-hydroxyl-oct-2,7-die-noate-2-*O*-β-d-glucopyranosyl-β-d-glucopyranoside (C_22_H_35_O_14_^−^). In addition, peak **26** with a parent ion [M − H]^−^ at *m/z* 453.19665 was resolved as hexamethyl-glucopyranuronosyl-glucoside methyl ester (C_19_H_33_O_12_^−^) and peak **31** with a [M − H]^−^ ion at *m/z* 393.17552 was determined as hexenyl-xylopyranosyl-glucose (C_17_H_29_O_10_^−^), peak **39** as secolonitoside (C_21_H_35_O_12_^−^). Finally, peak **59** with a parent ion [M − H]^−^ at *m/z* 421.20816 was resolved as oct-1-en-3-yl-arabinosyl-glucopyranoside (C_19_H_33_O_10_^−^) and peak **40** as Trehalose undecylenoate (C_23_H_39_O_12_^−^). Peak **19** was identified as linamarin.

### 2.2. Total Phenolics, Flavonoid Contents, and Antioxidant Activity of G. sphacelata

The in vitro results of total phenolics, total flavonoid content, and antioxidant activity are summarized in Table 2. DPPH, ABTS, FRAP, and superoxide anion scavenging activity (O_2_^−^) were used to measure the antioxidant potential in pulp and seed of *G. sphacelata* fruits. These in vitro assays are very simple and widely employed to calculate the antioxidant activities of plant and fruit extracts. The results were compared with previous studies from our research as well as work on other Chilean fruits by other researchers [20,21]. The total phenolic contents of pulp of Chilean berries from *G. sphacelata* (45.44 ± 0.67 mg GAE/g dry weight, Table 2) was closer to previous studies reported for the blueberries high-bush type *Vaccinium corimbosum* L (45.86 ± 3.46 mg GAE/g) [20] and Chilean blackberries maqui (*Aristotelia chilensis* (Molina) Stuntz) (49.74 ± 0.57 mg GAE/g) [22]. In addition, our TPC values were almost twice the values for TPC (29 mg Q/g) of the Chilean blackberries *Luma apiculata* (DC.) Burret (commonly known as arrayán or palo colorado) [13]. The amount of TFC found, 35.57 ± 0.86 mg QE/g, was lower than those of Chilean berries *Berberis microphilla* G. Forst. (45.72 ± 2.68 mg QE/g) commonly known as calafate or michay [20]. In the DPPH assay, the trapping capacity of *G. sphacelata* (487.11 ± 26.22 μmol Trolox/g dry weight) was close to that obtained for the red Chilean berries *Ugni molinae* Turcz. (commonly known as murtilla, murta, murtillo, or uñi), (450 μmol Trolox equivalents, TE/g dry weight) which is considered as an average antioxidant level in the category of edible fruits [21]. The ABTS values (190.32 ± 6.23 μmol TE/g dry weight) were lower than those of numerous Latin American fruits, such as Chilean blackberries maqui with a trapping capacity of 254.8 ± 8.2 μmol TE/g dry weight, and Brazilian Acai (*Euterpe oleraceae* Mart.), with 211.0 ± 14 μmol TE/g dry weight [23], while the FRAP values (169.08 ± 9.81 μmol TE/g dry weight) were close to those of Acai (157.9 ± 8.7 μmol TE/g dry weight) [23], but lower than maqui (254.2 ± 2.6 mg TE/g dry weight) [22,23]. The superoxide anion scavenging activity (O_2_^−^) of *G. sphacelata* (76.46 ± 3.18%) was higher compared to those established for the blueberries growing in Chile *Vaccinium corimbosum* L. (72.61 ± 1.91%) [20]. These values can classify these *G. sphacelata* fruits as moderate to high antioxidant small fruits such as plum, cherries, and strawberries [23].

### 2.3. In Vitro Cholinesterase Inhibitory Assay

The inhibition of key enzymes, such as AChE and BChE (linked to Alzheimer’s disease), is an important metric for identifying novel and safe medicinal value from natural products, especially those found in edible fruits. Indeed, four of the approved anti-Alzheimer drugs are cholinesterase inhibitors. For this reason, we tested the cholinesterase inhibitory potential of *G. sphacelata* against AChE and BChE. The results are shown in Table 3 and expressed as IC_50_ values. Galantamine was used as a positive control (0.27 ± 0.03 μg/mL against AChE and 3.82 ± 0.02 μg/mL against BChE). *G. sphacelata* fruits showed promising results for both enzymes. In the AChE assay, the inhibition IC_50_ for pulp was 4.49 + 0.08 μg/mL and for seeds was 4.38 ± 0.04 μg/mL. As for the BChE assay, the inhibition IC_50_ for pulp was 73.86 ± 0.09 μg/mL and for seeds was 78.57 ± 0.06 μg/mL. No significant difference was observed.

Some metabolites contained in *G. sphacelata* have been linked to counteraction against Alzheimer’s disease in previous reports. The iridoid jatamanvaltrate H isolated from *V. jatamansi* showed moderate neuroprotective activity [24,25]. In the same way, the crude extract of *V. jatamansi* and its fractions have shown considerable activity against AChE [26]. Marrubiin exhibited potent antinociceptive effects in a dose-dependent manner [27]. Congestiflorone acetates, such as detected in this study, have shown significant inhibitory effects against AChE with an IC_50_ value at 20.25 ± 0.55 µM [28]. On the other hand, the phenolic glucosyringic acid, did not show significant inhibitory effect on BChE (IC_50_ > 100 µM) [29]. Catechin presents neuroprotective effects as evidenced by amyloid β-induced neurotoxicity by MTT, lactate dehydrogenase (LDH) release, and neutral red uptake assays [30], but insignificant inhibitory against AChE [31]. Recently, the inhibitory activity against BChE of a procyanidin B1 was reported [32]. To the best of our knowledge, this is the first scientific report regarding the cholinesterase inhibitory potential of *G. sphacelata* fruits.

### 2.4. Docking Studies

In order to get insights on the intermolecular interactions, the most abundant compounds according to the UHPLC chromatogram (Figure 2) obtained from the pulp and seeds of *G. sphacelata* as well as the known cholinesterase inhibitor galantamine, were subjected to docking assays into the *Tc*AChE catalytic site and *h*BChE catalytic site. In order to rationalize their pharmacological results analyzing their protein molecular interactions in the light of their experimental inhibition, activities are shown in Table 3. The best docking binding energies expressed in kcal/mol of each compound are shown in Table 4.

#### 2.4.1. Acetylcholinesterase (TcAChE) Docking Results

Table 3 showed that the flavonoid quercetin-3-*O*-glucoside-acetate, and the isoflavones lupinisoflavone and genistein-7-*O*-di-glucoside displayed the best binding energies of −9.46, −9.36, and −9.18 kcal/mol, respectively. These results suggest that the *G. sphacelata* pulp or seed extracts inhibitory activity over acetylcholinesterase are mainly due the compounds mentioned above, especially the flavonoid quercetin-3-*O*-glucoside-acetate.

Pulp and seeds extract presented considerably abilities to exert an inhibitory potency over the *Tc*AChE enzyme (IC_50_ = 4.49 ± 0.08 for pulp extract and IC_50_ = 4.38 ± 0.04 for seeds extract) considering the known cholinesterase inhibitor galantamine (see Table 3). In this sense, Figure 3 shows the hydrogen bond interactions in a two-dimensional diagram of each main and most abundant compounds determined from both extracts into the *Tc*AChE catalytic site to summarize the information.

#### 2.4.2. Butyrylcholinesterase (hBuChE) Docking Results

All binding energies obtained from docking assays over butyrylcholinesterase (*h*BChE) of the most abundant compounds in the pulp and the seeds extracts were shown to be poorer compared to those in *Tc*AChE. These results are consistent with the less inhibitory activity of the extracts over this enzyme shown in Table 2 (IC_50_ = 73.86 ± 0.09 for pulp extract and IC_50_ = 78.57 ± 0.06 for seeds extract).

Just like in *Tc*AChE, the flavonoid quercetin-3-*O*-glucoside-acetate exhibited the best binding energy profile, suggesting that this derivative could be the main responsible for the inhibitory activity over the *h*BChE.

## 3. Materials and Methods

### 3.1. Chemicals and Plant Material

HPLC grade solvents (ethanol, formic acid, and acetonitrile) were obtained from Merck (Santiago, Chile). Ultrapure water was obtained from a water system of purification (Milli-Q Merck Millipore, Chile). Flavonol standards (catechin, isoflavones, and flavonoids, with a high purity: 95% by HPLC) were acquired from ChromaDex (Santa Ana, CA, USA), Sigma-Aldrich (Saint Louis, Mo, USA), or Extrasynthèse (Genay, France). Folin-Ciocalteu reagent, NaOH, Na_2_CO_3_, AlCl_3_, FeCl_3_, HCl, NaNO_2_, trichloroacetic acid, quercetin, Trolox sodium acetate, gallic acid, 2,4,6-tri(2-pyridyl)-s-triazine (TPTZ), nitroblue tetrazolium, xanthine oxidase, and DPPH (1,1-diphenyl-2-picrylhydrazyl radical), acetylcholinesterase (AChE) from electric eel (*Torpedo californica*), butylcholinesterase (BChE) from horse serum, 5,5′-dithiobis(2-nitrobenzoic) acid (DTNB), acetylthiocholine iodide, butyrylthiocholine chloride, and galantamine were purchased from Sigma-Aldrich Chemical Company (Santiago, Chile)*. G. sphacelata* fruits were collected in November 2016 in Valdivia, XIV Region de Los Ríos, Chile. The fruits were identified by Jorge Macaya (Universidad de Chile) and a voucher specimen were kept at the Institute of Pharmacy of the Universidad Austral de Chile under number GS20161115.

### 3.2. Fruit Processing

Five g of pulp and seeds, separately, were chopped and lyophilized (FreeZone 2.5 L Labconco, USA). The material was extracted (20% *w*/*v*) with a mixture of ethanol-distilled water (1:1 *v*/*v*) as solvent (at 25 °C, for two h in an ultrasonic bath). The extract was filtered, and the solvent was evaporated under vacuum at 45 °C. The extract was frozen and lyophilized, until a yield of 327.3 and 125.8 mg of dark brown gums (6.54 and 2.51%) from pulp and seeds, respectively.

### 3.3. UHPLC-PDA-Orbitrap-MS

The Dionex Thermo Scientific Ultimate 3000 UHPLC system connected with a Thermo Q Exactive Focus machine was used as previously informed [33]. Samples were re-dissolved (2 mg/mL) in ethanol-distilled water (1:1 *v*/*v*) and 10 µL of filtered solution (PTFE filter) were injected in the instrument, as previously discussed [33].

### 3.4. LC Parameters and MS Parameters

Liquid chromatography (LC) was executed by a C-18 Acclaim UHPLC column (×4.6 mm ID 150 mm, 2.5 µm, Thermo Fisher Scientific, Bremen, Germany) set at 25 °C. The wavelength detection was set at: 280, 354, 254, and 330 nm, and DAD was attained from 200 to 800 nm for full characterization of peaks. Mobile phases employed were acetonitrile (B) and 1% formic aqueous solution (A) while the gradient program was: (0.00 min, 5% B); (5.00 min, 5% B); (10.00 min, 30% B); (15.00 min, 30% B); (20.00 min, 70% B); (25.00 min, 70% B); (35.00 min, 5% B); and 12 min for column equilibration before injections. The flow rate employed was 1.00 mL min^−1^, and the injection volume was 10 µL. Standards and the fruit extracts dissolved in methanol were maintained at 10 °C during storage in the auto-sampler. The HESI II and Orbitrap spectrometer parameters were set as informed previously [34].

### 3.5. Total Phenolic (TP) and Total Flavonoid (TF) Content

The total phenolic contents and total flavonoid content of the *G. sphacelata* fruits were measured using the Folin-Ciocalteu and FeCl_3_ method previously described by our work with some modifications [35]. For TP, the results were expressed as mg of gallic acid equivalents per gram of dry fruit. While for the TF content results were presented as mg of quercetin equivalents per gram of dry fruit. The determination was performed using a Synergy HTX microplate reader (Biotek, Winoosky, VT, USA), in triplicate and reported as the mean ± SD.

### 3.6. Antioxidant Assays

#### 3.6.1. DPPH Cation Radical Discoloration Test

The method previously reported by Brand- Williams et al., was used to determine radical DPPH scavenging activity. Briefly, 2 mL of the DPPH solution was added in 400 μL of the extract (2 mg/mL) and mixed and 1.10 ± 0.02 at 517 nm absorbance was adjusted with methanol. This homogenized mixture of fruit extract and DPPH solution was then kept in a dark environment for a period of 20 min at room temperature [36]. Finally, the absorbance was calculated at 517 nm. The inhibition percentage was measured by a given formula:Percentage Inhibition = [1 − (S.A/B.A)] × 100(1)
where S.A. is sample absorbance and B.A. is used for blank absorbance.

#### 3.6.2. Bleaching Test with the Cationic Radical ABTS^•+^

The ABTS*^•^*^+^ radical capacity was evaluated using the decolorization method described by Kuskoski et al., 2004 [37]. The ABTS solution (7 mM) and potassium persulfate (2.45 mM) solution were prepared and mixed in a ratio of 1:1. The resultant solution was incubated for 16 h. Using 96-well plates, 275 µL of the solution (absorbance 0.7) was mixed with 25 µL of samples/standard. The absorbances were measured at 734 nm after 6 min of incubation period at 30 °C using a Synergy HTX microplate reader. Results were expressed as µmol Trolox per milliliters.

#### 3.6.3. Ferric Reduction Ability-Antioxidant Power Test (FRAP)

The FRAP test was performed with the previously described protocol by Benzie and Strain, with a slight modification [38]. Briefly, the FRAP solution (2 mL) mixed with 200 μL of extract (2 mg/mL), was stirred and kept in the dark for 5 min. The absorbance was measured at 595 nm.

#### 3.6.4. Superoxide Anion Scavenging Assay

The assay was carried using xanthine oxidase and hypoxanthine, and the absorbances were measured at 560 nm according to the reported with some modifications [13]. The production of the enzyme xanthine oxidase of superoxide anion radical (O_2_^−^) reduces the NBT (nitro blue tetrazolium) dye, producing a chromophore that absorbs at 520 nm. The superoxide anion trapping capacities of the extracts were measured using a Synergy HTX microplate reader (Biotek, CA, USA).

### 3.7. Cholinesterases (ChE) Inhibitory Activity

Ellman’s method was used to determinate the inhibitory activity of *G. sphacelata* fruits [39]. DTNB was dissolved in Tris HCl (pH 8) containing 0.02 M of MgCl_2_ and 0.1 M NaCl. The sample solution (50 mL dH_2_O, 2 mg L^−1^) was mixed with DTNB (125 mL), acetylcholinesterase (AChE; or BChE) (25 mL), while the blank sample as a control contained all the solutions except enzymes and was distributed in a 96-well microplate. The acetyl-thiocholine iodide (ATCI) or butyryl-thiocholine chloride (BTCl) (25 mL) was added to start the reaction. The reading was taken on a 405 nm absorbance after incubation at 25 °C for 15 min. The cholinesterase inhibitory activity was measured as IC_50_ (μg mL^−1^, concentration range 0.05 to 25 μg mL^−1^) by subtracting the absorbance of the sample from blank. Galantamine was used as a positive control. All data were collected in triplicate. In addition, 0.26 units/mL of each enzyme were used for the inhibition assays.

### 3.8. Docking Studies

The geometries and partial charges of several representative compounds contained in the extracts, as well as the known cholinesterases (*Tc*AChE- *h*BChE) inhibitor Galantamine were fully optimized using the DFT method with the standard basis set PBEPBE/6-311 + g* [40,41]. All calculations were performed in the Gaussian 09W software.

### 3.9. Statistical Analysis

All the experiments were repeated five times to confirm the results and minimize the error and the data was presented as the mean of standard deviation. The results were analyzed using one-way analysis of variance (ANOVA) and Tuckey test statistical analysis (*p*-values < 0.05 were regarded as significant) using the origin Pro 9.0 software package (Origin lab Corporation, Northampton, MA, USA).

## 4. Conclusions

From the edible endemic Chilean *G. sphacelata* fruit seventy metabolites were detected using UHPLC-PDA-Orbitrap-MS analysis including phenolic acids, organic acids, sugar derivatives, catechins, proanthocyanidins, fatty acids, iridoids, coumarins, a benzophenone, flavonoids, and terpenes. *G. sphacelata* showed a good phenolic content and moderate antioxidant and enzyme inhibitory activities. This report could contribute for the better understanding of chemistry and biological activities in the genus *Greigia*. Furthermore, the list of compounds profiled with a cholinesterase inhibitory activity plus antioxidant potential make *G. sphacelata* fruits an interesting source of compounds with interesting properties to prepare functional foods or food derived supplements.

## Figures and Tables

**Figure 1 molecules-25-03750-f001:**
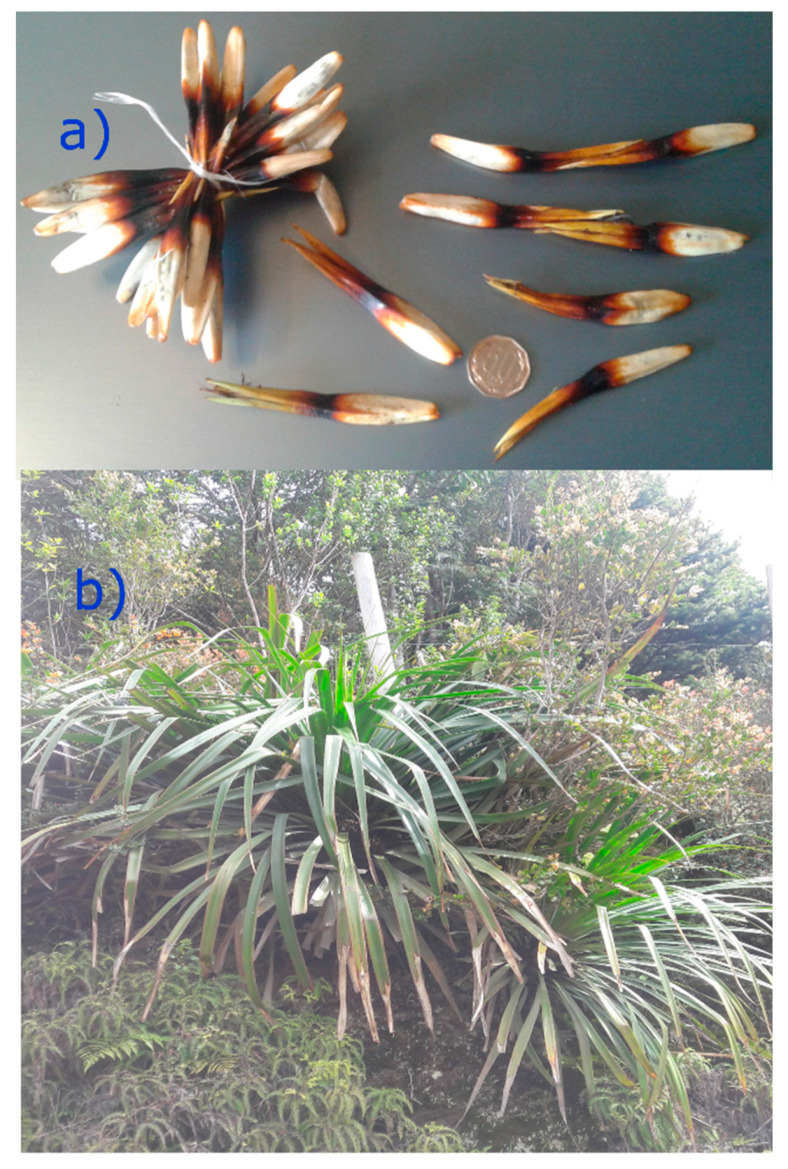
*Greigia sphacelata* (Ruiz and Pav.) Regel (*Bromeliaceae*). (**a**) Fruits, (**b**) plant.

**Figure 2 molecules-25-03750-f002:**
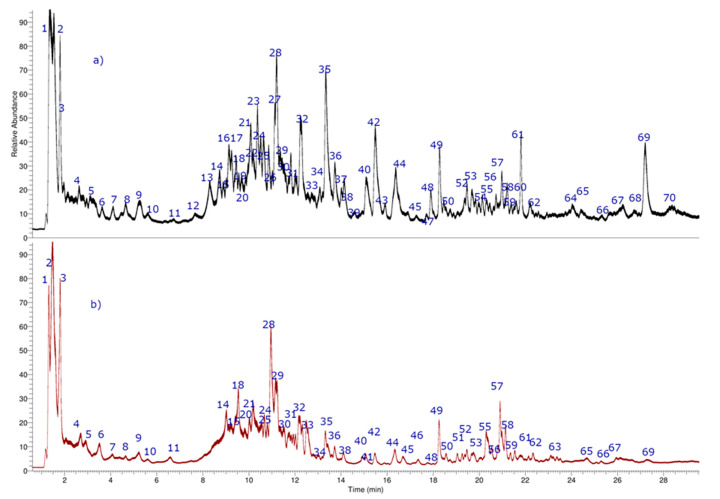
UHPLC chromatograms of *G. sphacelata* (**a**) pulp, and (**b**) seeds.

**Figure 3 molecules-25-03750-f003:**
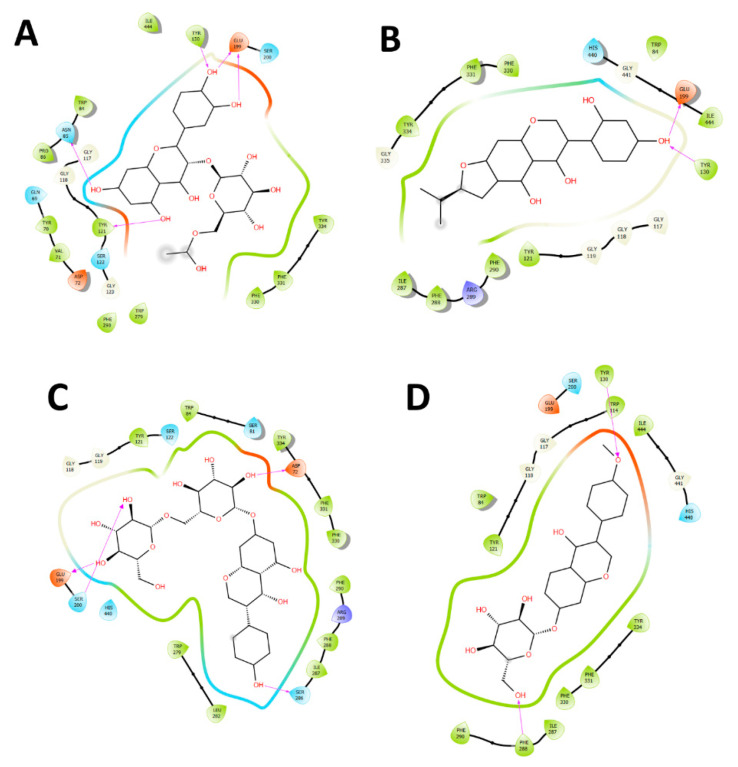
Two-dimensional diagram of *Torpedo californica* acetylcholinesterase (*Tc*AChE) catalytic site and hydrogen bond interactions of each main and most abundant compounds obtained from the pulp and seeds extracts. (**A**) Quercetin-3-*O*-glucoside-acetate (flavonoid); (**B**) Lupinisoflavone (isoflavone); (**C**) Genistein-7-*O*-di-glucoside (isoflavone); (**D**) Ononin (formononetin 7-*O*-glucoside) (isoflavone); (**E**) Genistein-7-*O*-glucoside (isoflavone); (**F**) Aesculetin-7-*O*-glucuronide (coumarin); (**G**) Dihydroxy-octadecaenoic acid (fatty acid); (**H**) Hydroxy-pentadecanoic acid (fatty acid).

**Table 1 molecules-25-03750-t001:** Full metabolome identification in *Greigia sphacelata* by UHPLC-PDA-Orbitrap-MS.

Peak	Tentative Identification	[M − H]^−^ Ions	Retention Time (min)	UV Max (nm)	Theoretical Mass (*m/z*)	Measured Mass (*m/z*)	Accuracy (ppm)	Metabolite Type	MS^2^ Ions (ppm)	Fruit Part
1	Syringaldehyde syringate	C_18_H_17_O_9_^−^	1.36	315	377.08578	377.08671	−2.44	Pa	215.03258	P; S
2	Quinic acid	C_7_H_11_O_6_^−^	1.45		191.05501	191.05621	3.16	Oa	135.02919	P; S
3	Citric acid	C_6_H_7_O_7_^−^	1.52	187	191.01863	191.01915	3.12	Oa	162.89121, 111.00782	P; S
4	Dehydroascorbic acid	C_6_H_5_O_6_^−^	2.20	176	173.00806	173.00879	4.18	Oa		P; S
5	Isocitric acid	C_6_H_7_O_7_^−^	2.74		191.01863	191.01923	3.76	Oa	162.89110, 111.00810	P; S
6	Diffutidin	C_17_H_18_O_5_^−^	3.45		301.10705	301.10614	−3.02	Pa	153.01875	P; S
7	Homocitric acid	C_10_H_17_O_4_^−^	4.05		205.03542	205.03491	−2.43	Oa		P; S
8	Vanilloyl-glucoside	C_14_H_17_O_9_^−^	4.34	287	329.08810	329.08671	4.24	Pa	151.03874, 123.02314	P; S
9	Euonyminol	C_15_H_25_O_10_^−^	5.26		365.14422	365.14560	3.76	Te		P; S
10	Pantothenic acid	C_9_H_16_NO_5_^−^	5.255		218.10230	218.10313	3.82	Oa	125.85231	P; S
11	Ebuloside	C_21_H_31_O_10_^−^	6.50	175	443.19263	443.19117	3.27	Ir		P; S
12	Nonioside k	C_17_H_29_O_11_^−^	7.85		409.17144	409.17191	3.58	Sd		P
13	Glucosyringic acid	C_15_H_19_O_10_^−^	8.56	325	359.09841	359.09824	-0.47	Pa	197.0455 (syringic acid)	P
14	Acetylshikonin	C_18_H_17_O_6_^−^	9.01	305	329.10162	329.10114	−0.47	Pa	171.07063	P; S
15	Allyl-sucrose	C_15_H_25_O_11_^−^	9.12		381.14023	381.14066	2.86	Sd		P; S
16	Feruloyl-*O*-galactarate	C_16_H_17_O_11_^−^	9.25	246–310	385.07797	385.07654	3.64	Pa	194.0577, 133.8732	P
17	Cinnamoyl catalpol	C_24_H_27_O_11_^−^	9.29	198	491.15589	491.15479	−1.74	Ir	175.06085	P
18	Cinnamoyl catalpol isomer	C_16_H_21_O_9_^−^	9.49	198	357.11801	357.11938	−1.74	Ir	258.09830	P; S
19	Linamarin	C_10_H_15_NO_6_^−^	9.65		246.09821	246.09825	4.21	Sd	258.09830	P
20	Amurensin	C_26_H_29_O_12_^−^	10.03	202–284	533.16431	533.16425	−1.96	Fl	503.14331, 341.1015	P; S
21	2-caffeoylisocitric acid	C_15_H_13_O_10_^−^	9.98		353.07032	353.05173	3.97	Pa	179.06425, 154.99808	P; S
22	Methyl 2,3,4-tris-*O*-[2-(carboxymethyl) ethyl]- glucose derivative	C_26_H_31_O_13_^−^	10.12		479.17592	479.17749	3.27	Sd		P
23	Methyl 2,3,4-tris-*O*-[2-(carboxymethyl) ethyl]-glucose	C_19_H_31_O_12_^−^	10.43		451.18228	451.18100	2.83	Sd		P
24	Dimethyl--hydroxyl-oct-2,7-die- noate-glucopyranosyl-glucose	C_22_H_35_O_14_^−^	10.55		523.20213	523.20331	2.24	Sd	241.07164, 125.02764	P; S
25	Aesculetin-7-*O*-glucuronide	C_15_H_13_O_10_^−^	10.62	232–279–329	353.05140	353.05130	-0.28	Co	177.0193 (Aesculetin)	P; S
26	Hexamethyl-glucopiranuronosyl-glucoside methyl ester	C_19_H_33_O_12_^−^	10.83		453.19800	453.19794	2.96	Sd		P
27	7-Hydroxycoumarin glucuronide	C_15_H_13_O_9_^−^	10.95	255–355	337.05541	337.05661	3.56	Co		P
28	Quercetin-3-*O*-glucoside-acetate	C_23_H_21_O_13_^−^	11.03	255–355	505.09767	505.09912	2.87	Fl	301.02755 (Quercetin)	P; S
29	Tectoridin	C_22_H_21_O_11_^−^	11.23	281	461.10894	461.10941	3.39	Fl; Is	300.2632 (tectorigenin)	P; S
30	Escopoletin 7-*O*-glucuronide	C_16_H_15_O_10_^−^	11.35	320	367.06597	367.06729	3.59	Co	171.06593	P; S
31	Hexenyl-xylopiranosyl-glucose	C_17_H_29_O_10_^−^	11.71		393.17706	393.17552	3.91	Sd		P; S
32	Lupinisoflavone A	C_16_H_15_O_9_^−^	12.12	285	351.07245	351.07106	3.96	Fl; Is	321.0763, 241.0502	P; S
33	2,2,3-Tris (2,3-dihydroxypropanoyl) decanoic acid	C_19_H_31_O_11_^−^	12.43		435.18609	435.18750	3.23	Fa	287.15002	P; S
34	Trimeric proanthocyanidin C2	C_45_H_37_O_18_^−^	12.96	283	865.19859	865.19451	−4.71	Ca	577.1342, (dimer), 289.0714 catechin, (C_15_H_14_O_6_^−^)	P; S
35	Genistein-7-*O*-di-glucoside	C_27_H_29_O_15_^−^	13.20	230–238–333	593.15122	593.15033	−1.65	Fl; Is	269.0452 (genistein)	P; S
36	Genistein-7-*O*-di-galactoside	C_27_H_29_O_15_^−^	13.52	230–238–333	593.15120	593.15033	−1.63	Fl; Is	269.0451 (genistein)	P; S
37	Procyanidin B1	C_30_H_23_O_12_^−^	13.83	285	577.13445	577.13373	1.16	Ca	289.0714 catechin, (C_15_H_14_O_6_^−^)	P
38	Genistein-7-*O*-glucoside	C_21_H_19_O_10_^−^	14.03	230–238–333	431.09847	431.09824	−0.53	Fl; Is	269.0452 (genistein)	P; S
39	Secolonitoside	C_21_H_35_O_12_^−^	14.35		479.21230	479.21350	2.50	Sd		P
40	Trehalose undecylenoate	C_23_H_39_O_12_^−^	15.13		507.24360	507.24454	1.84	Sd	457.06708	P; S
41	Proanthocyanin tetramer	C_60_H_49_O_24_^−^	15.32	285	1153.26191	1153.25146	−9.05	Ca	577.1337 (dimer)	S
42	Bis-2-hydroxyethyl phtalate	C_12_H_13_O_6_^−^	15.52	315	253.07173	253.07066	4.21	Pa		P; S
43	Daidzein-7-*O*-galactoside	C_21_H_19_O_9_^−^	15.63	281	415.10236	415.10327	3.00	Fl; Is	253.05065	P
44	Procyanidin A1	C_30_H_23_O_12_^−^	16.45	282	575.11958	575.11877	−1.42	Ca	289.07157 (catechin)	P; Se
45	Daidzein-7-*O*-glucoside	C_21_H_19_O_9_^−^	16.75	230–285–326	415.10236	415.10361	3.00	Fl; Is	273.04050	P; S
46	Ononin (formononetin 7-*O*-glucoside)	C_22_H_21_O_9_^−^	17.26	220–285–326	429.11917	429.11880	−0.86	Fl; Is	267.06601, 251.0362 (formononetin)	P
47	1-*O*-*trans*-*p*-coumaroylglycerol	C_12_H_13_O_6_^−^	17.84	212–326	253.07184	253.07150	−1.34	Pa	119.04922, 120.05251	S
48	1, 3-*O*-di-*trans*-*p*-coumaroylglycerol	C_21_H_19_O_9_^−^	18.03	212–326	415.10355	415.10333	2.86	Pa	119.04920, 120.05248	P; S
49	Catechin*	C_15_H_13_O_6_^−^	18.32	285	289.07162	289.07164	0.06	Fl; Ca	260.06575, 245.08210	P; S
50	Tetrahydroxy-tetradecadienoic acid	C_14_H_23_O_6_^−^	18.45	265	287.15012	287.15001	4.50	Fa		P; S
51	Monic acid A	C_17_H_27_O_7_^−^	19.03	215	343.17513	343.17657	4.21	Te		P; S
52	Pentahydroxy-octadecatetranoic acid	C_18_H_27_O_7_^−^	19.47	189	355.17514	355.17648	3.81	Fa		Pu; Se
53	Dictamnoside N	C_18_H_29_O_8_^−^	19.75	175	373.18569	373.18707	3.69	Te	217.88326	P; S
54	Evodinnol	C_14_H_16_O_4_^−^	19.83	325	247.09649	247.09760	4.49	Pa	133.02882	P
55	Dihydroxy-pentadecanoic acid	C_15_H_29_O_4_^−^	20.36	215	273.20724	273.20734	0.36	Fa		P; S
56	12-hydroxyoctadecanoic acid	C_18_H_36_O_3_^−^	20.52	225	299.25946	299.25807	4.64	Fa		P; S
57	Congestiflorone	C_28_H_31_O_4_^−^	20.98	243	431.22169	431.22061	−2.49	Be	365.17944, 269.17584, 152.99516	P; S
58	Hydroxy-pentadecaenoic acid	C_15_H_27_O_4_^−^	21.06	234	271.19144	271.19173	1.06	Fa		P; S
59	Oct-1-en-3-yl-arabinosyl-glucopyranoside	C_19_H_33_O_10_^−^	21.57		421.20821	421.20816	−0.11	Sa		P; S
60	Marrubiin	C_20_H_27_O_4_^−^	21.73	225	331.19196	331.19039	4.73	Te	207.19521	P
61	Quillaic acid	C_30_H_45_O_5_^−^	21.85	214	485.32788	485.32615	3.56	Te	405.32125, 249.03424	P; S
62	Dihydroxy-octadecaenoic acid	C_18_H_33_O_4_^−^	22.05	225	313.23883	313.23734	−4.76	Fa		P; S
63	Pentadecanedioic acid	C_15_H_27_O_4_^−^	23.25	235	271.19307	271.19165	4.66	Fa		S
64	Trihydroxy-octadecadienoic acid	C_18_H_31_O_5_^−^	24.21	235	327.21770	327.21780	0.30	Fa		P
65	Trihydroxy-octadecaenoic acid	C_18_H_33_O_5_^−^	24.68	222	329.23332	329.23341	0.27	Fa		P; S
66	2,3-dihydroxypropyl dodecanoate	C_15_H_29_O_4_^−^	25.36	198	273.20715	273.20724	4.42	Fa		P; S
67	Tetrahydroxy-heptadecatrienoic acid	C_17_H_27_O_6_^−^	26.03	285	327.1813	327.18142	3.89	Fa		P; S
68	Jatamanvaltrate H	C_22_H_33_O_9_^−^	26.19	215	441.21335	441.21329	3.26	Ir	389.62607, 311.22302, 135.04443	P
69	Hydroxy-pentadecanoic acid	C_15_H_29_O_3_^−^	27.45	235	257.21221	257.21222	0.03	Fa		P; S
70	Unknown	C_14_H_29_O_8_^−^	28.33	220	325.18569	325.18442	−3.92	Fa		P

Metabolite type: Be: Benzophenones; Ca: Catechins; Co: Coumarins; Fa: Fatty acids; Fl: Flavonoids; Ir: Iridoids; Is: Isoflavones; Oa: Organic acids; Pa: Phenolic acids; Sd: Sugar derivatives; and Tr: Terpenes. Fruit part: P: Pulp and S: Seed.

**Table 2 molecules-25-03750-t002:** Total phenolic contents, total flavonoid contents, and antioxidant activities of *G. sphacelata.*

Sample	TPC ^a^	TFC ^b^	DPPH ^c^	ABTS ^d^	FRAP ^e^	O_2_^− f^ (%)
Pulp	45.44 ± 0.67	35.57 ± 0.86	487.11 ± 26.22	190.32 ± 6.23	169.08 ± 9.81	76.46 ± 3.18 ^b^
Seeds	37.21 ± 0.45	28.32 ± 0.35	354.51 ± 34.16	140.49 ± 3.58	147.84 ± 4.35	67.02 ± 2.23 ^b^

^a^ Total phenolic content (TPC), expressed in mg of gallic acid equivalents per gram of dry plant. ^b^ Total flavonoid content (TFC), expressed in mg of quercetin equivalents per gram of dry plant. ^c^ 1,1-diphenyl-2-picrylhydrazyl radical (DPPH), as μmol of Trolox equivalent/g dry fruit, ^d^ ABTS as μmol of Trolox equivalent/g dry fruit. ^e^ Ferric reducing antioxidant power (FRAP), as μmol TE/g dry weight. ^f^ Superoxide anion scavenging activity (O_2_^−^) expressed in %. All values were expressed as means ± SD (*n* = 3). The results are statistically compared with a positive control. The results were analyzed using one-way analysis of variance (ANOVA) and Tuckey test statistical analysis (*p*-values < 0.05 were regarded as significant). Values in the same column marked with the same letter are not significantly different (at *p* < 0.05).

**Table 3 molecules-25-03750-t003:** Cholinesterase inhibitory activity of pulp and seed of *G. sphacelata.*

Sample	AChE (IC_50_)	BChE (IC_50_)
Pulp	4.49 ± 0.08	73.86 ± 0.09
Seeds	4.38 ± 0.04	78.57 ± 0.06
Galantamine	0.27 ± 0.03	3.82 ± 0.02

The results are compared with their positive control. Values correspond to the average of three experiments; all values were expressed as means ± SD. Units of concentrations are expressed as (μg/mL). The results were analyzed using one-way analysis of variance (ANOVA) and Tuckey test statistical analysis (*p*-values < 0.05 were regarded as significant).

**Table 4 molecules-25-03750-t004:** Binding energies obtained from docking experiments of most abundant compounds in *G. sphacelata*’s fruit and the known cholinesterase inhibitor galantamine over acetylcholinesterase (*Tc*AChE) and butyrylcholinesterase (*h*BChE).

Compound	Binding Energy (kcal/mol) Acetylcholinesterase (*Tc*AChE)	Binding Energy (kcal/mol) Butyrylcholinesterase (*h*BChE)
Quercetin-3-*O*-glucoside-acetate	−9.46	−8.31
Lupinisoflavone	−9.36	−7.99
Genistein-7-*O-*di-glucoside	−9.18	−6.89
Ononin (formononetin 7-*O*-glucoside)	−7.45	−6.44
Genistein-7-*O*-glucoside	−7.24	−5.86
Aesculetin-7-*O*-glucuronide	−6.67	−6.85
Dihydroxy-octadecaenoic acid	−4.71	−5.76
Hydroxy-pentadecanoic acid	−4.81	−4.87
Galantamine	−11.81	−9.5

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
