# Peer review of "Chemical Fingerprinting and Biological Evaluation of the Endemic Chilean Fruit Greigia sphacelata (Ruiz and Pav.) Regel (Bromeliaceae) by UHPLC-PDA-Orbitrap-Mass Spectrometry"

_molecules, 2020, doi:10.3390/molecules25163750_

Round 1
Reviewer 1 Report
The current paper characterizes the potential bioactives and biological activity of the native Chilean fruit Greigia sphacelata. The authors employ HPLC-PDA-Orbitrap-MS to nicely characterize a wide range of phytochemicals within the fruit. The authors then looked at potential bioactivity through several in vitro measures of antioxidant activity, and cholinesterase inhibitory potential. While the dataset is of value for the initial characterization of this fruit, the manuscript suffers from being poorly written in parts; particularly with English language use, and being too wordy and repetitive in parts. Moreover, the authors fail to account in their use comparative examples with other fruits that in vitro activity may not translate to in vivo response. Therefore, the paper could use specific examples of the potential role of the most abundant compounds in in vivo models. Phenolics and polyphenols such as the flavonoids vary considerably in their bioavailability, metabolism and ultimately biological effect. While antioxidant activity assays are of value to give a quick and rough comparative estimate of the phenols/polyphenols present in a food compared to others, it can never be a substitute for a food or food components potential to affect health with intake.
Additional minor comments:
In the results and discussion section there are separate sections for flavonoids, isoflavones, and catechins and proanthocyanidins. However, catechins, proanthocyanidins and isoflavones are flavonoids.
Reference 56 needs to be corrected.
Author Response
We thank the reviewer for his valuable comments. We plan to make in vivo experiments in the near future. We have corrected the manuscript. Yes. Of course, antioxidant activity assays just provide a comparative estimate of the phenols/polyphenols present in a food compared to others, it can never be a substitute for a food or food components potential to affect health with intake. However, without trying, how do we know our materials is not as good as others are? We are the first to look at the seed material and investigate it’s in vitro antioxidant activity. There are a lot of previous related studies that use the same in vitro results to compare their efficacy of the material as we try to do. We also regret to see that our result did not make a soaring high AChE and BChE activates but it was the fact we got and presented.
Additional minor comments:
In the results and discussion section there are separate sectionsfor flavonoids, isoflavones, and catechins andproanthocyanidins.
However, catechins, proanthocyanidins and isoflavones are flavonoids.
R: corrected
Reference 56 needs to be corrected
R: corrected
Reviewer 2 Report
The authors did a lot of tedious analytical work, which deserves to be published. However, the part of study dealing with the AChE is confusing and not really necessary, especially because the conclusions in respect to the AD cannot be done according to the findings presented. Therefore, omitting the respective parts of the study/manuscript will increase its soundness, while a rather general statement about possible beneficiary effects of the fruits of the particular plant for the integrative biomedical approach to oxidative stress-associated disorders will be more appropriate.
Author Response
Thank you so much for your nice comments and suggestions. We understand that the results of AChE and BChE are not very promising that can reflect the potential source for the AD disease. However, the results presented in this study should be considered as the first information on the anticholinesterase activity of Chilean fruit Greigia sphacelata. The finding may be suggestive for future work related to neurodegenerative disorder. Moreover, some investigations suggest that radical scavenging activity was found to be related to anticholinesterase activity. Therefore, in this case it will be good to mention both the results in our manuscript. We will highly appreciate if you consider these results in the manuscript.
We thank the reviewer for his opinions however, we did a lot of work trying to improve this manuscript, we spent money and time doing these experiments, also co-author Javier Romero Parra made the calculations using Gaussian and wrote the conclusions so we prefer to keep this part in this manuscript. The manuscript was improved with this measurement of AChE activities.
Reviewer 3 Report
The study entitled “Chemical fingerprinting and bioactivity of the Native Antioxidant Chilean Fruit Greigia sphacelata (Ruiz & Pav.) Regel (Bromeliaceae) by UHPLC-HESI-Orbitrap-Tandem-Mass Spectrometry” describes the identification of secondary metabolites in the extract of pulp and seeds of fruits of Greigia sphacelata using Ultrahigh Performance Liquid Chromatography. In addition, starting from this extract, the authors investigate antioxidant properties (Trolox Equivalents, DPPHF, ABTS) and enzyme inhibition of acetylcholinesterase (AChE) and butyrylcholinesterase (BChE). Finally, a docking study is performed for interactions between representative compounds and AChE/BChE. Cholinesterase inhibitors are commonly used to prevent the depletion of the neurotransmitter acetylcholine in some neurodegenerative diseases, e.g. Alzheimer’s disease.
The chemical characterization of G. sphacelate fruits is novel and the assays were performed with sufficient number of independent replicates (n=3-5). The most bioactive compounds against acetylcholine were not inequivocally identified, nevertheless the authors present a docking study identifying promising compounds.
Concerning specific aspects of the study presented, some points need to be addressed by the authors:
1: Introduction (lines 60-65) - when discussing metabolites found in several parts of G. sphacelata, the authors cite Donno et al 2015 (reference 11); however, that study does not mention this plant species.
2: Introduction – the introduction ends with a technical consideration on UPHLC-MS without any statement on the experimental hypothesis or goals of the study. In general, the context of the study should be explicit in the last part of the introduction.
3: Results and Discussion – (subsection 2.1.13.) – acetylcholine is given the same abbreviation as acetylcholinesterase (AChE); the abbreviation for acetylcholine needs to be removed or changed to a common abbreviation (e.g. ACh). In addition, the work of Ahmad et al (reference 40) does not discuss the role of ACh as neurotransmitter and its effect on memory, unlike suggested in the manuscript.
4: Results and Discussion – Table 2 – all the bioactivity assays are collected in one table; however, for reasons of clarity, it would be preferable to have the antioxidant part separated from the cholinesterase assays, especially since there are controls (e.g. the ChE inhibitor galanthamine) that are omitted from the table and from statistical treatment.
5: Results and Discussion – 3.4 Docking studies – in one part of the study butyrylcholinesterase is abbreviated as BChE, but in this section it becomes BuChE; uniformization is required.
6: Results and Discussion – 3.4 Docking studies – why Torpedo californica acetylcholinesterase but human BChE were chosen for the docking study?
7: Materials and Methods – subsections 4.1 and 4.6.5.1 – Essential information is missing - what is the biological origin of the AChE and BChE used, are they both human? From where were these enzymes purchased? How many units of each enzyme were used for the ChE inhibition assays?
8: Materials and Methods – 2.6.7 statistical analysis – also from table 2 what are the groups that are statistically compared? Pulp extract vs. seed extract? Are there comparisons with positive controls? This should be more clearly defined in the manuscript.
Author Response
The introduction is too long. Shorten it.
R: We have cut unnecessary content and it greatly reduced the length of Introduction. We hope it will satisfy the reviewer.
It is very strange that no examples of new compounds or unidentified compounds were observed. For example, in Fig.1 the peak between 1 and 2 was left unidentified. Peaks 19 and 22 are double. Peak 31 have shoulder. Please provide more information and characterisation of such cases. It is not enough just to match the obtained spectra with the database.
R: Thank you so much to the detailed analyses of the reviewer. Such complex mixtures it is difficult to get all information. At least this is new approach to an unstudied new fruit. We have revised again the table and added some more peaks.
In Table 1 are calculated wrong. The exact masses of the compounds are a bit off. Entry 2 and 4 have a different theoretical mass even though they are isomeric and isobaric compounds. This is impossible. Please correct all theoretical masses.
R: Corrected.
Table 2: Units in DPPH are mangled in my copy of the manuscript. Also the authors state: “All values were expressed as means + SD (n = 3). The data are presented as the mean ± standard error for five determinations of each test (n= 5). Were the experiments performed 3 or 5 times?
R: We are extremely sorry for this inconvenience as it was a typos error. We critically revised the manuscript for these kinds of errors and corrected. All the experiments including DPPH were performed in triplet and the data is presented as the mean ± standard error for three determinations of each test (n= 3).
Reviewer 4 Report
The present paper deals with the chemical fingerprinting of Chilean fruit Greigia sphacelata using orbitrap mass spectrometry.
Suggestions:
The introduction is too long. Shorten it.
It is very strange that no examples of new compounds or unidentified compounds were observed. For example in Fig.1 the peak between 1 and 2 was left unidentified. Peaks 19 and 22 are double. Peak 31 have shoulder. Please provide more information and characterisation of such cases. It is not enough just to match the obtained spectra with the database.
In Table 1 are calculated wrong. The exact masses of the compounds are a bit off. Entry 2 and 4 have a different theoretical mass even though they are isomeric and isobaric compounds. This is impossible. Please correct all theoretical masses.
Table 2: Units in DPPH are mangled in my copy of the manuscript. Also the authors state: “All values were expressed as means + SD (n = 3). The data are presented as the mean ± standard error for five determinations of each test (n= 5). Were the experiments performed
3 or 5 times?
Author Response
The study entitled “Chemical fingerprinting and bioactivity of the Native Antioxidant Chilean Fruit Greigia sphacelata (Ruiz & Pav.) Regel (Bromeliaceae) by UHPLC-HESI-Orbitrap-Tandem-Mass Spectrometry” describes the identification of secondary metabolites in the extract of pulp and seeds of fruits of Greigia sphacelata using Ultrahigh Performance LiquidChromatography. In addition, starting from this extract, the authors investigate antioxidant properties (Trolox Equivalents, DPPHF, ABTS) and enzyme inhibition of acetylcholinesterase(AChE) and butyrylcholinesterase (BChE). Finally, a docking study is performed for interactions between representative compounds and AChE/BChE. Cholinesterase inhibitors are commonly used to prevent the depletion of the neurotransmitter acetylcholine in some neurodegenerative diseases, e.g. Alzheimer’s disease.
The chemical characterization of G. sphacelate fruits is novel and the assays were performed with sufficient number of independent replicates (n=3-5). The most bioactive compounds against acetylcholine were not inequivocally identified, nevertheless the authors present a docking study identifying promising compounds. Concerning specific aspects of the study presented, some points need to be addressed by the authors:
1: Introduction (lines 60-65) - when discussing metabolites found in several parts of G. sphacelata, the authors cite Donno et al 2015 (reference 11); however, that study does not mention this plant species.
R: Deleted
2: Introduction – the introduction ends with a technical consideration on UPHLC-MS without any statement on the experimental hypothesis or goals of the study. In general, the context of the study should be explicit in the last part of the introduction.
R: Corrected
3: Results and Discussion – (subsection 2.1.13.) – acetylcholine is given the same abbreviation as acetylcholinesterase (AChE); the abbreviation for acetylcholine needs to be removed or changed to a common abbreviation (e.g. ACh). In addition, the work of Ahmad et al (reference 40) does not discuss the role of ACh as neurotransmitter and its effect on memory, unlike suggested in the manuscript.
R: Updated
4: Results and Discussion – Table 2 – all the bioactivity assays are collected in one table; however, for reasons of clarity, it would be preferable to have the antioxidant part separated from the cholinesterase assays, especially since there are controls (e.g. the AChE inhibitor galanthamine) that are omitted from the table and from statistical treatment.
R: Repaired
5: Results and Discussion – 3.4 Docking studies – in one part of the study butyrylcholinesterase is abbreviated as BChE, but in this section it becomes BuChE; uniformization is required.
6: Results and Discussion – 3.4 Docking studies – why Torpedo californica acetylcholinesterase but human BChE were chosen for the docking study?
R: Electric eel Torpedo californica is the most common enzyme used.
7: Materials and Methods – subsections 4.1 and 4.6.5.1 – Essential information is missing - what is the biological origin of the AChE and BChE used, are they both human? From where were these enzymes purchased? How many units of each enzyme were used for the AChE inhibition assays?
R: We have incorporated the required information in the section 4.1 and 4.6.5.1.
8: Materials and Methods – 2.6.7 statistical analysis – also from table 2 what are the groups that are statistically compared? Pulp extract vs. seed extract? Are there comparisons with positive controls? This should be more clearly defined in the manuscript.
R: All the results are compared with their positive control. The new changes were incorporated in the new section 3.7
Round 2
Reviewer 1 Report
Thank you to the authors for the work they have done on the revision, it has been greatly improved, and the comparison examples provided in the antioxidant activity section are of value. However, yes there are a lot of studies that “use the same in vitro results to compare their efficacy of the material as we try to do”, however, ultimately, most of the components found in any food are not going to be bioavailable to any extent, and will actually be considerably changed structurally once absorbed. A good example are the glucosides used in the docking experiments. These if consumed will be cleaved by beta-glucosidases in the gut or transformed by the gut microbiota to either the parent aglycone or other unique compounds. So that is why this reviewer asks what the ultimate goal of this research, is it to define a functional food, supplement or is to identify potential bioactive compounds that future work will be needed to determine how to deliver them safely and intact to the target? The literature is unfortunately full of examples of in vitro studies that cannot be translated to the in vivo condition. With that said it is of considerable value to this review to fully identify the compounds in any food, as the author’s did for Greigia sphacelata, and I applaud the authors for that.
Some minor corrections are still needed in terms of grammar and content:
Abstract:
2nd line: change “and produce” to “that produces”
4th to the last line, remove the line, "From the findings, promising results were observed for pulp and seeds.” And add to the beginning of the next sentence, “the pulp and seeds from” after the words “suggest that”
Last line: need to add “potential” in front of interest
Introduction:
2nd to last line in the 2nd paragraph from the bottom of page: Please add “a” in front of “key”
Last sentence of Intro: move “for the first time” to after “report”.
Section 2.2 The examples inserted are helpful to add context
Not certain why the authors want to separate out the catechins and proanthocyanidins from the other flavonoids, but I suggest making it more clear that flavanols are also flavonoids. It’s important as the 6 flavonoid subclasses are routinely examined in epidemiology studies with regards to their intake and chronic disease risk. Recent examples can be found: https://doi.org /10.1080/10408398.2018.1476964; https://doi.org/10.1007/s00394-020-02218-z
Section 2.1.4 – Please provide the reference that describes oxylipins as “dietary antioxidants”. At the minimum they are bioactive lipids that are important in a plant’s response to stress. In mammals, they are predominately from polyunsaturated fatty acid metabolism so not certain how much dietary intake will contribute to their bioactivity if at all.
First line of page 16: Please confirm that the three iridoids detected all have the described bioactivities. If not change the sentence structure to describe iridoids as a class as having the described bioactivities.
First line of page 18: Please add the reference(s) that support the statement.